# SIMPLIFY IN-CONTEXT LEARNING

## ABSTRACT

Traditional in-context learning (ICL) enhances the performance and capability of large language models (LLMs) primarily by optimizing decomposition strategies, reformatting, and ordering. However, when task difficulty significantly exceeds the model's capabilities, merely refining in-contexts becomes ineffective. In contrast to prior work that focuses on improving the capabilities of LLMs, we propose Simplified In-context Learning (SICL), a framework that reduces task difficulty through task decomposition. A complex task $A$ is decomposed into a sequence of subtasks $A_1, A_2, \cdots, A_m$, each less difficult than the original. When the difficulty of a subtask $A_i$ lies within the capability threshold of the LLM, LLM still achieves strong performance. SICL incorporates two complementary strategies: training-free methods that achieve rapid decomposition through clustering and uniform partitioning of the output space; and training-driven methods that adaptively determine the optimal decomposition for each query via a scoring predictor. Empirically, SICL achieves state-of-the-art (SOTA) results across six tasks, three LLMs and ten datasets, attaining the lowest mean squared error (MSE) of 0.712 and the highest accuracy of 77.0%. We further extend SICL to generative tasks, where it achieves a ROUGE-1 of 0.270 on summarization and a BLEU of 0.254 on machine translation. Furthermore, SICL also generalizes to the vision modality, yielding a maximum accuracy of 98.8% on image classification. Notably, SICL demonstrates consistent SOTA performance over commercial LLMs, like GPT-4o.

## 1 INTRODUCTION

Large language models (LLMs) demonstrate remarkable performance across a wide range of language tasks (Bertsch et al., 2025; Nafar et al., 2025). In-context learning (ICL) is a prominent paradigm that enhances the capabilities of LLMs by leveraging demonstrations to construct an in-context (Jin et al., 2025; Ren et al., 2025). Typically, ICL retrieves relevant demonstrations from a retrieval dataset and concatenates the task description, demonstrations (in-context), and input text as input to the LLM (Wen et al., 2025; Wang et al., 2023). Owing to its training-free nature and strong empirical effectiveness, ICL is a widely adopted technique for LLMs (Zhang et al., 2025).

Previous ICL methods primarily focus on optimizing the retrieval of in-context demonstrations (Gao et al., 2024; Zhao et al., 2025), refining their formatting (Li et al., 2024b; Kassianik et al., 2025), and determining their optimal order (Cao et al., 2025; Wang et al., 2025). These studies aim to improve LLM performance by enhancing how demonstrations are retrieved and organized. When an LLM's capability exceeds the difficulty of a task, it delivers strong performance. For example, as shown in the subfig. (a) of Fig. 1, if an LLM's capability is 6 and the task's difficulty is 8, traditional ICL methods attempt to increase the LLM's capability to 10, thereby surpassing the task difficulty and improving performance. Although these methods achieve promising results, they mainly focus on augmenting the LLM's capability while neglecting the inherent difficulty of the task itself.

However, the performance improvement afforded by in-context learning (ICL) is intrinsically bounded. Specifically, when the complexity of a task surpasses the aggregate of the large language model's (LLM) inherent capacity and the marginal gain provided by ICL, conventional ICL approaches inevitably yield suboptimal performance. To address this limitation, we propose **Simplified In-Context Learning (SICL)**, which shifts the focus from enhancing model capability to reducing task difficulty. By simplifying the original task, SICL ensures that the LLM's capability exceeds the task difficulty, thereby enabling strong performance. Specifically, we decompose the original task $A$ into a series of subtasks $A_1, A_2, \cdots, A_m$, where each subtask $A_i$ is less difficult

than the original task $A$. For example, as shown in the subfig. (b) of Fig. 1, a task with a difficulty level of 8 can be decomposed into four subtasks with difficulties of 1, 1, 2, and 4, while the total difficulty remains 8 and LLM's capability 6. In this case, the LLM's capability exceeds the difficulty of each subtask, thereby facilitating effective performance.

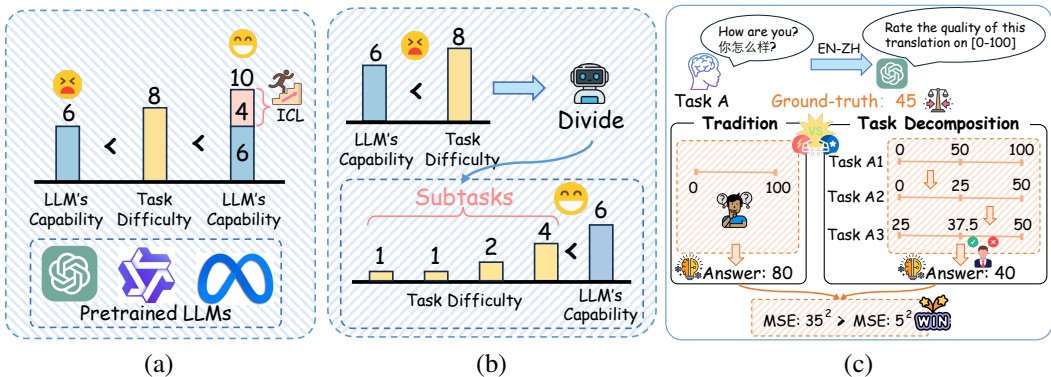

(a)  (b)  (c)

Figure 1: (a) Traditional ICL improves performance by retrieving relevant demonstrations to boost effective capability above the task difficulty. (b) SICL decomposes the task into simpler subtasks, ensuring the LLM's capability exceeds each subtask's difficulty. (c) An example of reducing task difficulty by progressively constraining the output space through subtasks.

A key challenge in task decomposition lies in ensuring that each subtask is less difficult than the original task. We establish that coarse-grained labels (e.g., "dog") are inherently easier to predict than fine-grained labels (e.g., "Corgi"), as they encompass broader categories and are associated with a reduced output space containing fewer distracting alternatives. Consequently, our method reduces overall task difficulty by sequentially constraining the output space via coarse-grained labels, capitalizing on its hierarchical structure to facilitate decomposition. For example, as shown in the subfig. (c) of Fig. 1, in a translation quality evaluation task, an LLM with in-context decomposition assigns a score between 0 and 100 to a translated sentence. Predicting an exact score within such a broad range is challenging. To address this issue, we decompose the task into three subtasks. The first subtask, $A_1$, requires the LLM to determine whether the score falls within 0−50 ([0,50]) or 50−100 ([50,100]). If the prediction is 0−50 ([0,50]), the second subtask, $A_2$, asks whether the score lies within 0−25 ([0,25]) or 25−50 ([25,50]). If the second prediction is 25−50 ([25,50]), the third subtask, $A_3$, requires the LLM to assign a score within this narrower range. Compared with predicting a score across the full 0−100 ([0,100]) range, the first and second subtasks involve only binary decisions, making them simpler. The third subtask further narrows the range from 0−100 ([0,100]) to 25−50 ([25,50]), thereby reducing the difficulty. Through this form of decomposition, we reduce task complexity and difficulty, enabling LLMs to generate more accurate results.

As illustrated in Figure 2, the process requires determining both the width and depth of the decomposed subtasks. **Width** denotes the number of labels within each subtask. For instance, task $A_1$

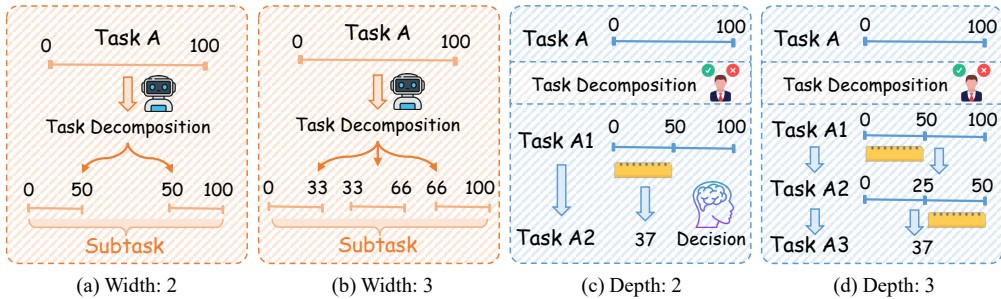

(a) Width: 2  (b) Width: 3  (c) Depth: 2  (d) Depth: 3

Figure 2: Illustration of Task Width and Depth. The width of a subtask is defined as its number of labels, whereas the depth is defined as the number of constituent subtasks.

may be framed as a binary classification (e.g., Label 1: 0−50; Label 2: 51−100) or as a ternary classification (e.g., Label 1: 0−33; Label 2: 33−66; Label 3: 66−100). Hence, the number of labels defines the task's width. **Depth**, in contrast, refers to the number of subtasks produced through decomposition. In the preceding example, the original task is divided into 3 subtasks, yielding a depth of 3. Alternatively, it could be decomposed into only 2 subtasks—for instance, after predicting whether the label lies within [0, 50] or [50, 100], the model can directly estimate the final label value. Thus, the challenge of SICL can be reformulated into the following questions:

① How should the number of labels in each decomposed subtask (i.e., the decomposition width) be determined? ② How should the number of subtasks (i.e., the decomposition depth) be determined?

SICL addresses these questions through two complementary strategies. The **training-free method** fixes the decomposition depth at two and sets the first-layer subtask width to two. The **training-driven method** partitions a subset of the dataset to train a score predictor, which estimates LLM performance under different width–depth configurations for a given input. The configuration with the greatest predicted score is selected as the optimal decomposition. This enables SICL to adaptively tailor both the width and depth of subtasks to each query, improving performance across tasks.

We evaluate SICL on three LLMs, ten benchmark datasets, and six tasks, achieving state-of-the-art (SOTA) performance across all settings. Specifically, SICL reaches 77.0% accuracy in text classification, 0.712 MSE in semantic textual similarity, and 188.858 MSE in translation quality evaluation. For generation tasks, SICL achieves a ROUGE-1 score of 0.140 on summarization, a ROUGE-1 score of 0.270 on text expansion, and a BLEU score of 0.254 on machine translation. SICL proves applicable to the vision modality, reaching a maximum accuracy of 98.8% on image classification. In addition, it demonstrates SOTA performance on proprietary commercial models. Our contributions are summarized as follows: ❶ We formalize the notion that reducing task difficulty enhances the effectiveness of in-context learning. ❷ We propose Simplified In-Context Learning (SICL), a framework designed to reduce task difficulty, and present both training-free and training-driven strategies tailored for a wide range of downstream tasks. ❸ We empirically validate the effectiveness of SICL on six downstream tasks and ten benchmark datasets, achieving SOTA performance.

## 2    RELATED WORK

Demonstration organization critically impacts ICL performance. Prior work explores selection, reformatting, and ordering. Selection includes unsupervised methods retrieving top-$k$ neighbors by similarity, mutual information, perplexity, or model probabilities. (Sorensen et al., 2022; Nguyen & Wong, 2023; Li & Qiu, 2023b; Wang et al., 2025), and supervised approaches, which employ task-specific training and adaptive strategies (Rubin et al., 2022; Li et al., 2023; Ye et al., 2023; Mavromatis et al., 2023). Most methods adopt a fixed $k$, though its task-specific justification remains underexplored (Li et al., 2023; Qin et al., 2023; Oorloff et al., 2025). Reformatting enhances alignment with LLMs via self-generated examples, structured prompting, or representation-level adaptations (Kim et al., 2022; Liu et al., 2024a; Li et al., 2024a). Ordering addresses sequence sensitivity using entropy, similarity-based proximity, or curriculum-based ranking (Liu et al., 2022a; 2024c). *Current ICL emphasizes enhancing LLM capability via in-context adjustments, but neglects task difficulty reduction, an area demanding urgent research attention.*

## 3    METHOD

SICL can be divided into two main components. The first is a demonstration-retrieval method that constructs in-context prompts by selecting demonstrations from a retrieval dataset according to their similarity to the input text. The second is a task decomposition method that partitions the original task into subtasks to reduce its difficulty.

### 3.1    CONSTRUCTION PROCESS OF IN-CONTEXT

In the in-context construction phase, following KATE Liu et al. (2022b) and ICL-rerank Zhou et al. (2024), we retrieve the top-$k$ demonstrations from the retrieval dataset $\mathbf{D}_r$ that are most semantically aligned with the input text. These are then used to construct the in-context for the query. Specifically, the retrieval process consists of three main steps: **(1) Encoding Representations**: Each input text

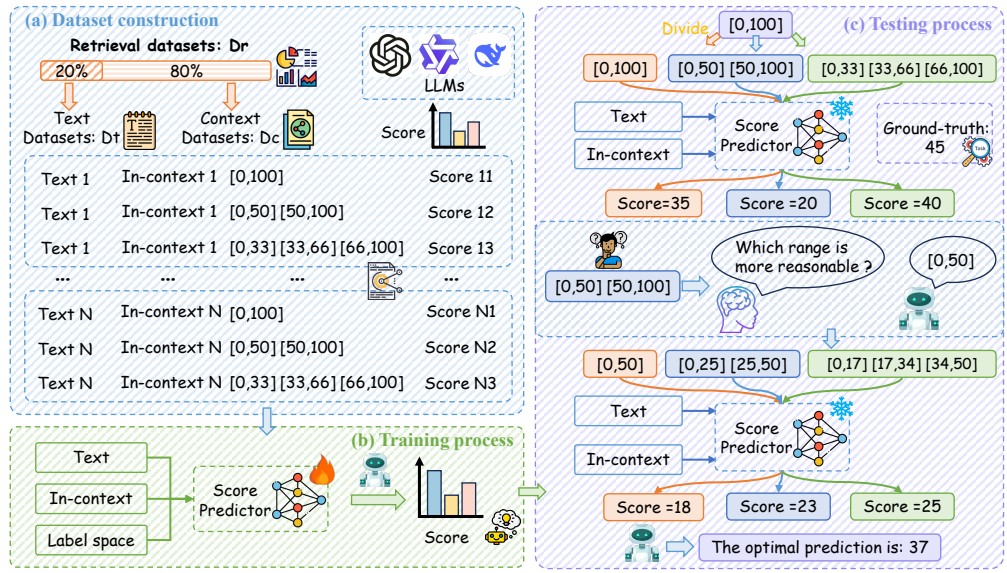

Figure 3: Overall framework of train-driven simplified in-context learning (SICL).

$x_i^{\text{te}} \in \mathbf{D}\text{t}$ from the test dataset and each candidate demonstration $x_j^{\text{r}} \in \mathbf{D}\text{r}$ from the retrieval dataset are mapped into dense vectors using a pre-trained encoder $f_{\text{pre}}$. Specifically, $\mathbf{e}_i^{\text{te}} = f_{\text{pre}}(x_i^{\text{te}})$, $\mathbf{e}_j^{\text{r}} = f_{\text{pre}}(x_j^{\text{r}})$, and $\mathbf{E} = \{\mathbf{e}_1^{\text{r}}, \cdots, \mathbf{e}_h^{\text{r}}\}$, where $h$ denotes the size of the retrieval dataset and $j \leq h$. **(2) Measuring Semantic Proximity**: Cosine similarity is used to compute the similarity score $s_{ij}$ between $\mathbf{e}_i^{\text{te}}$ and $\mathbf{e}_j^{\text{r}}$, i.e., $s_{ij} = \frac{\mathbf{e}_i^{\text{te}} \cdot \mathbf{e}_j^{\text{r}}}{\|\mathbf{e}_i^{\text{te}}\| \cdot \|\mathbf{e}_j^{\text{r}}\|}$, and the similarity profile of $x_i^{\text{te}}$ is $\mathbf{S}_i = \{s_{i1}, \ldots, s_{ih}\}$. **(3) In-context Construction**: The top-$k$ pairs with the highest similarity are selected and arranged in descending order to construct the in-context $\mathbf{D}_i^{\text{in,te}}$ for $x_i^{\text{te}}$, i.e., $\mathbf{D}_i^{\text{in,te}} = (x_w^{\text{in}}, y_w^{\text{in}})_{w=1}^{k}$, where $x_w^{\text{in}}$ denotes the $w$-th most similar demonstration from $\mathbf{D}\text{r}$ to $x_i^{\text{te}}$, and $y_w^{\text{in}}$ is the label of $x_w^{\text{in}}$.

## 3.2 TASK DECOMPOSITION METHOD

As outlined in Section 1, decomposing the original task into subtasks of lower difficulty requires determining both the depth of the decomposition and the width of each subtask. To address this, we introduce two approaches. The first is a training-free decomposition method that can be implemented efficiently; we set the task depth to 2 and fix the width of the first subtask at 2. The second approach trains a `Score Predictor` to adaptively determine the depth and width of subtasks for each input text. The details of these methods are presented below.

**1. Training-free decomposition method-SICL (train-free):** In the training-free decomposition method, we set the subtask depth to 2 and fix the first subtask width at 2. For classification tasks with discrete labels, the first subtask's label space is split into two groups using K-means (Xu & Tian, 2025) clustering algorithm. For regression tasks with continuous scores, the label space of the first subtask is evenly divided at the midpoint to form two corresponding label subsets. Formally,

$$\mathcal{Y}^{(1)} = \begin{cases} \{C_1, C_2\}, & \text{if classification (via K-means clustering)} \\ \{[y_{\min}, \frac{y_{\min} + y_{\max}}{2}], \ [\frac{y_{\min} + y_{\max}}{2}, y_{\max}]\}, & \text{if regression (split at midpoint)} \end{cases} \tag{1}$$

where $\mathcal{Y}^{(1)}$ denotes the label space of the first subtask, $C_1$ and $C_2$ are the two clusters obtained by K-means, and $y\min$ and $y_{\max}$ denote the minimum and maximum values of the label space.

**2. Training-driven decomposition method-SICL (train-driven):** Just as prescribing a uniform medical dosage disregards individual differences in patients, imposing a fixed configuration of subtask depth and width neglects the inherent variability of query texts. Because each input is unique, a static setting in the training-free decomposition method cannot be assumed to be universally optimal. This observation suggests that adaptively determining subtask depth and width for individual

input text is more likely to yield effective results. Therefore, we introduce **Score-Prediction (SP)** framework that evaluates LLM performance under different subtask configurations and adaptively selects the one with the greatest predicted score for each input. As shown in Fig. 3, (text, in-context, label space, score) tuples are constructed to train a deep learning model (Score-Prediction) with three inputs and one output (score). During inference, subtask depth is progressively increased from 1 to its maximum, and for each depth, the width with the greatest predicted performance score is selected. The overall procedure consists of three stages: ❶ training dataset construction, ❷ model training, and ❸ progressive selection of subtask depth and width.

**2.1 Training dataset construction:** Building on the preceding analysis, we construct the training data for Score Predictor as tuples of the form (text, in-context, label space, score). Here, the score denotes an evaluation metric of the LLM's output when a given text is processed with a specified in-context and label space. Computing this metric requires both predicted and ground-truth values. In the ICL setting, ground-truth values are only available in the retrieval dataset. Therefore, we partition the retrieval dataset into a text dataset for training and a context dataset for retrieval. Specifically, as Fig. 3 shows, the retrieval dataset $\mathbf{Dr}$ is randomly divided into a text dataset $\mathbf{Dt}$ (20%) and a context dataset $\mathbf{Dc}$ (80%). Formally,

$$\mathbf{Dr} = \mathbf{Dt} \cup \mathbf{Dc}, \quad \mathbf{Dt} \cap \mathbf{Dc} = \varnothing, \quad \mathbf{Dt} \sim \text{RandomSample}(\mathbf{Dr}, 20\%), \quad \mathbf{Dc} = \mathbf{Dr} \setminus \mathbf{Dt}. \quad (2)$$

For each text $x_i^t \in \mathbf{Dt}$, its corresponding in-context $\mathbf{D}_i^{\text{in,t}}$ are retrieved from the context dataset $\mathbf{Dc}$ using the method described in Section 3.1.

After identifying the training texts and their corresponding in-contexts (text, in-context), we construct the associated (label space, score). For the label space, the maximum subtask width is defined as $w$ (reduced to $l$ if the number of labels $l$ in the subtask label space is smaller than $w$), and the maximum subtask depth is denoted as $d$. For classification tasks, in the first subtask, the label space is clustered into $w$ groups. For regression tasks with score outputs, the score range is evenly divided into $w$ intervals. Formally, let $\mathcal{Y}$ denote the original label space with cardinality $|\mathcal{Y}| = l$. Given a maximum width $w$ and depth $d$, the effective width is defined as $w^* = \min(w, l)$. The label space of the first subtask is then expressed as

$$\mathcal{Y}^{(1)} = \begin{cases} \{C_1, C_2, \ldots, C_{w^*}\}, & \text{classification task,} \\ \{I_1, I_2, \ldots, I_{w^*}\}, & \text{regression task,} \end{cases} \quad (3)$$

where $I_j = \left[ y_{\min} + \frac{(j-1)}{w^*}(y_{\max} - y_{\min}), \ y_{\min} + \frac{j}{w^*}(y_{\max} - y_{\min}) \right), \quad j = 1, \ldots, w^*$. $C_i$ denotes the $i$-th cluster obtained from $\mathcal{Y}$ by K-means, and $y_{\min}, y_{\max}$ denotes the minimum and maximum values of the regression label range.

Then, as illustrated in Fig. 4, the text, in-context, and label space are provided to the LLM, which first predicts the interval of the label into which the text falls. Within the interval chosen by the LLM, the model subsequently estimates the final output of the original task. For instance in Fig. 4, in the translation quality evaluation task with a scoring range of [0, 100], the LLM initially decides whether the score lies in [0, 50] or [50, 100]. If [0, 50] is selected, the LLM then predicts the final score within this interval. After generating predictions, the outputs are compared with the ground-truth labels to compute the corresponding scores. Formally, given a text $x_i^t$, its in-context $\mathbf{D}_i^{\text{in,t}}$, and a label space $\mathcal{Y}$, the LLM $f_{\text{LLM}}$ generates a prediction $\hat{y}_i^t$, which is compared with the ground-truth label $y_i^t$ to compute the score:

$$\hat{y}_i^t = f_{\text{LLM}}(x_i^t, \mathbf{D}_i^{\text{in,t}}, \mathcal{Y}), \qquad \text{Score}(x_i^t, \mathbf{D}_i^{\text{in,t}}, \mathcal{Y}) = Eval(\hat{y}_i^t, y_i^t), \quad (4)$$

where $Eval(\cdot, \cdot)$ denotes the evaluation metric. For regression tasks with continuous outputs, the score is defined as the mean squared error (MSE) between the predicted and ground-truth values. For classification tasks with categorical outputs, the score equals 1 if the predicted label matches the ground truth and 0 otherwise. Specifically, the score is defined as

$$\text{Score}(x_i^t, \mathbf{D}_i^{\text{in,t}}, \mathcal{Y}) = \begin{cases} (y_i^t - \hat{y}_i^t)^2, & \text{regression task (mean squared error),} \\ \mathbb{I}(\hat{y}_i^t = y_i^t), & \text{classification task (indicator function: 1 if correct, 0 otherwise),} \end{cases} \quad (5)$$

where $\hat{y}$ denotes the prediction, $y$ is the ground-truth label, and $\mathbb{I}(\cdot)$ is the indicator function.

**Model Training** In the model training process, we employ a three-input, single-output architecture for Score Predictor (SP). One branch processes the input text, while the other handles the combined in-context and label space; the output corresponds to the predicted performance score for each pair. The complete architecture and training procedure are described in Section 4.1.

**Progressive selection of subtask depth and width:** After training SP, the task decomposition for each test text under a maximum subtask width $w$ and depth $d$ involves up to $w^d$ candidate partitions, which is computationally intractable. To reduce complexity, we employ a greedy strategy: at each step, SP is used to evaluate candidate partitions of the current label space, and the partition with the greatest score is selected as the next subtask. The process terminates once no finer partition improves the predicted score. For example, as the testing process in Fig. 3 shows, when predicting a score in `[0,100]` with a maximum width of 3, the SP first estimates the score (MSE) for candidate partitions: `[0,100]`, `[0,50]`,`[50,100]`,

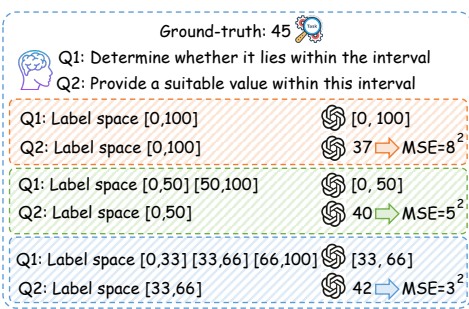

Figure 4: Example of score generation in translation quality evaluation, where the score is the MSE between the LLM's prediction and the ground truth.

and `[0,33]`,`[33,66]`,`[66,100]`. The partition with the greatest score (the lowest MSE) is chosen as the first subtask's label space. Then `[0,50]`,`[50,100]` is selected; the LLM then determines whether the score lies in `[0,50]` or `[50,100]`. If `[0,50]` is chosen, the SP again evaluates partitions of this interval—(`[0,50]`), (`[0,25]`,`[25,50]`), and (`[0,17]`,`[17,34]`,`[34,50]`)—and the best partition defines the second subtask. This process illustrates the greedy strategy for determining subtask widths. Subtask depth is determined in the same manner. If the current label space already achieves the best predicted score (e.g., minimal MSE), further splitting is unnecessary. For instance, in the second subtask, the SP predicts MSE values of `18`, `23`, and `25` for (`[0,50]`), (`[0,25]`,`[25,50]`), and (`[0,17]`,`[17,34]`,`[34,50]`), respectively, then `[0,50]` is retained without further division, and the LLM directly predicts within this interval. Formally, let $\mathcal{Y}$ denote the current label space, and $Eval(\hat{y}, y)$ be the evaluation metric (e.g., MSE). The greedy decomposition is defined as

$$g(\mathcal{Y}) = \begin{cases} \mathcal{Y}, & \min_{P \in \text{Partitions}(\mathcal{Y},w)} \text{SP}(P) \geq \text{SP}(\mathcal{Y}), \\ \arg\min_{P \in \text{Partitions}(\mathcal{Y},w)} \text{SP}(P), & \text{otherwise}, \end{cases} \tag{6}$$

where $\text{Partitions}(\mathcal{Y}, w)$ denotes the set of candidate partitions of $\mathcal{Y}$ with width at most $w$, and $\text{SP}(P)$ is the score predictor's estimated error for partition $P$. The function $g(\cdot)$ is applied recursively until depth $d$ is reached or further splitting is unnecessary.

**Extending train-driven SICL to generation tasks:** In the preceding sections, we focused on simplifying classification (label outputs) and regression (score outputs) tasks. Extending SICL to generation is more challenging due to the infinite output space. To address this, we propose a variant of SICL for generation tasks: ❶ The LLM first produces a set of top-$k$ candidate outputs. ❷ A score predictor is then trained to evaluate the quality of each candidate, and the candidate with the highest score is selected as the final output. Formally, let $x$ denote the input text and $\{\hat{y}_1, \hat{y}_2, \cdots, \hat{y}_k\}$ represent the top-$k$ candidates returned by $f_{\text{LLM}}(x)$. The final output is determined as:

$$\{\hat{y}_1, \hat{y}_2, \cdots, \hat{y}_k\} = \text{Top-}k\big(f_{\text{LLM}}(x)\big), \quad \hat{y} = \arg\max_{\hat{y}_i \in \{\hat{y}_1, \cdots, \hat{y}_k\}} g(\hat{y}_i), \tag{7}$$

where $\hat{y}$ denotes the final prediction for $x$, and $g(\cdot)$ is the score predictor.

# 4 EXPERIMENTS

## 4.1 EXPERIMENTAL SETUP

**Tasks, datasets and metrics:** We evaluate SICL on six NLP tasks across ten datasets: (1) Text Classification (SST-5, Emotion); (2) Semantic Textual Similarity (STSB, STS14); (3) Translation

Table 1: The results of SICL and other baselines. Best results in **bold**; second-best underlined.

| LLMs | Llama3.2 3B | | Llama3.1 8B | | Qwen2.5 7B | | Llama3.2 3B | | Llama3.1 8B | | Qwen2.5 7B | |
|---|---|---|---|---|---|---|---|---|---|---|---|---|
| Shot | 5 | 10 | 5 | 10 | 5 | 10 | 5 | 10 | 5 | 10 | 5 | 10 |
| Metric | MSE ↓ | | | | | | MSE ↓ | | | | | |
| Task & Data | Semantic Textual Similarity: STS14 | | | | | | Semantic Textual Similarity: STSB | | | | | |
| BM25 | 2.424 | 2.192 | 1.594 | 1.404 | 0.833 | 0.797 | 2.363 | 2.168 | 1.280 | 1.146 | 0.825 | 0.763 |
| CD | 3.928 | 2.831 | 2.333 | 1.263 | 1.097 | 0.965 | 5.271 | 2.785 | 1.498 | 1.291 | 0.878 | 0.898 |
| Kate | 2.301 | 2.198 | 1.663 | 1.441 | 0.850 | 0.813 | 2.285 | 2.033 | 1.188 | 1.138 | 0.807 | 0.747 |
| DKNN | 2.921 | 2.787 | 1.721 | 1.511 | 0.900 | 0.804 | 3.008 | 2.843 | 1.496 | 1.425 | 0.877 | 0.781 |
| TTF | 3.553 | 3.041 | 2.072 | 1.578 | 0.859 | 0.748 | 2.762 | 2.664 | 1.241 | 1.583 | 0.978 | 0.982 |
| ICCL | 2.392 | 2.252 | 1.492 | 1.361 | 0.803 | 0.727 | 2.382 | 2.321 | 1.167 | 1.199 | 0.793 | 0.786 |
| PPL | 2.113 | 2.538 | 1.986 | 1.372 | 0.973 | 0.989 | 2.587 | 2.528 | 1.783 | 1.403 | 0.830 | 0.816 |
| SICL (train-free) | 2.095 | 1.826 | 1.456 | 1.204 | 0.854 | 0.777 | 2.243 | 2.020 | 1.312 | 1.116 | 0.781 | 0.744 |
| SICL (train-driven) | **1.657** | **1.491** | **1.239** | **1.018** | **0.771** | **0.714** | **1.872** | **1.663** | **0.996** | **1.035** | **0.776** | **0.712** |
| Metric | MSE ↓ | | | | | | MSE ↓ | | | | | |
| Task & Data | Translation Quality Assessment: En-Ja | | | | | | Translation Quality Assessment: En-Zh | | | | | |
| BM25 | 261.930 | 260.993 | 302.528 | 291.224 | 213.164 | 223.592 | 337.338 | 337.205 | 348.522 | 369.705 | 264.243 | 259.650 |
| CD | 283.325 | 270.978 | 291.302 | 253.546 | 211.727 | 206.446 | 380.703 | 347.954 | 340.971 | 358.220 | 245.971 | 282.211 |
| Kate | 252.788 | 248.324 | 289.811 | 274.059 | 201.347 | 205.906 | 328.612 | 330.016 | 338.792 | 360.321 | 246.131 | 246.681 |
| DKNN | 312.839 | 296.596 | 282.790 | 290.201 | 219.362 | 212.360 | 373.765 | 359.280 | 350.190 | 361.087 | 266.456 | 264.404 |
| TTF | 255.171 | 288.628 | 274.893 | 273.374 | 205.480 | 200.481 | 343.487 | 384.456 | 346.975 | 308.749 | 247.978 | 262.553 |
| ICCL | 248.238 | 239.688 | 262.655 | 254.772 | 206.900 | 205.121 | 323.317 | 327.989 | 324.764 | 340.483 | 240.758 | 246.084 |
| PPL | 291.126 | 324.154 | 298.419 | 249.778 | 204.821 | 201.759 | 385.258 | 354.107 | 376.910 | 383.357 | 259.081 | 270.320 |
| SICL (train-free) | 221.206 | 228.918 | 260.910 | 228.619 | 201.521 | 204.252 | 263.979 | 283.531 | 335.380 | 324.609 | 250.886 | 238.510 |
| SICL (train-driven) | **207.729** | **205.910** | **224.510** | **224.273** | **189.606** | **188.858** | **251.718** | **251.771** | **268.240** | **260.052** | **236.253** | **221.814** |
| Metric | MSE ↓ | | | | | | ROUGE-1 ↑ | | | | | |
| Task & Data | Translation Quality Assessment: EN-CS | | | | | | Summarization: Gigaword | | | | | |
| BM25 | 4113.790 | 4094.444 | 541.510 | 531.930 | 345.527 | 352.005 | 0.022 | 0.022 | 0.022 | 0.020 | 0.128 | 0.108 |
| CD | 455.861 | 458.114 | 447.836 | 686.084 | 340.429 | 354.231 | 0.057 | 0.023 | 0.038 | 0.025 | 0.136 | 0.134 |
| Kate | 579.108 | 480.117 | 509.275 | 499.811 | 338.709 | 343.921 | 0.058 | 0.041 | 0.047 | 0.032 | 0.123 | 0.106 |
| DKNN | 534.045 | 531.965 | 487.242 | 517.100 | 362.391 | 354.981 | 0.021 | 0.025 | 0.004 | 0.028 | 0.126 | 0.114 |
| TTF | 536.601 | 642.720 | 397.321 | 666.192 | 460.978 | 401.761 | 0.028 | 0.054 | 0.037 | 0.036 | 0.127 | 0.114 |
| ICCL | 460.650 | 462.507 | 462.751 | 484.441 | 342.600 | 348.676 | 0.057 | 0.034 | 0.047 | 0.032 | 0.123 | 0.106 |
| PPL | 450.911 | 518.350 | 442.146 | 462.403 | 340.679 | 347.495 | 0.027 | 0.056 | 0.037 | 0.037 | 0.128 | 0.115 |
| SICL (train-free) | 417.896 | 457.192 | 390.562 | 476.836 | 348.662 | 341.793 | - | - | - | - | - | - |
| SICL (train-driven) | **348.134** | **387.913** | **348.753** | **392.051** | **326.825** | **329.477** | **0.078** | **0.073** | **0.048** | **0.043** | **0.139** | **0.140** |
| Metric | ROUGE-1 ↑ | | | | | | BLEU ↑ | | | | | |
| Task & Data | Text expansion: Gigatiny | | | | | | Machine Translation: WNT19-En-Zh | | | | | |
| BM25 | 0.129 | **0.132** | 0.129 | 0.128 | 0.134 | 0.135 | 0.003 | 0.002 | 0.034 | 0.033 | 0.040 | 0.040 |
| CD | 0.102 | 0.100 | 0.107 | 0.102 | 0.265 | 0.240 | 0.018 | 0.005 | 0.147 | 0.134 | 0.190 | 0.225 |
| Kate | 0.122 | 0.122 | 0.129 | 0.125 | 0.257 | 0.253 | 0.062 | 0.051 | 0.131 | 0.137 | 0.127 | 0.104 |
| DKNN | 0.111 | 0.096 | 0.108 | 0.107 | 0.257 | 0.237 | 0.001 | 0.001 | 0.122 | 0.138 | 0.152 | 0.174 |
| TTF | 0.087 | 0.087 | 0.100 | 0.099 | 0.245 | 0.264 | 0.001 | 0.001 | 0.133 | 0.146 | 0.023 | 0.008 |
| ICCL | 0.120 | 0.116 | 0.124 | 0.115 | 0.241 | 0.230 | 0.003 | 0.002 | 0.134 | 0.139 | 0.129 | 0.111 |
| PPL | 0.091 | 0.074 | 0.191 | 0.110 | 0.252 | 0.249 | 0.003 | 0.002 | 0.173 | 0.144 | 0.219 | 0.165 |
| SICL (train-free) | - | - | - | - | - | - | - | - | - | - | - | - |
| SICL (train-driven) | **0.160** | 0.129 | **0.140** | **0.135** | **0.270** | **0.269** | **0.071** | **0.068** | **0.218** | **0.223** | **0.249** | **0.254** |
| Metric | Accuracy (%) ↑ | | | | | | Accuracy (%) ↑ | | | | | |
| Task & Data | Text classification: Emotion | | | | | | Text classification: SST5 | | | | | |
| BM25 | 42.5 | 47.3 | 34.0 | 38.0 | 57.8 | 59.0 | 21.7 | 22.6 | 34.6 | 31.4 | 48.4 | 49.9 |
| CD | 23.8 | 18.0 | 43.3 | 39.8 | 54.0 | 55.5 | 10.0 | 13.3 | 28.7 | 36.4 | 37.6 | 45.9 |
| Kate | 45.0 | 48.5 | 43.8 | 40.0 | 71.8 | 72.3 | 17.2 | 17.6 | 35.5 | 35.7 | 49.5 | 51.8 |
| DKNN | 17.0 | 16.5 | 41.8 | 46.5 | 40.5 | 43.0 | 7.9 | 13.3 | 38.4 | 35.7 | 49.1 | 45.0 |
| TTF | 29.9 | 30.3 | 42.4 | 38.5 | 44.0 | 52.9 | 17.1 | 17.1 | 42.7 | 38.5 | 46.2 | 46.8 |
| ICCL | 43.6 | 48.0 | 42.0 | 38.2 | 69.0 | 69.7 | 19.2 | 19.7 | 38.2 | 31.7 | 50.7 | 48.8 |
| PPL | 42.0 | 37.9 | 40.1 | 39.4 | 52.6 | 51.0 | 13.6 | 17.5 | 32.7 | 32.1 | 49.0 | 48.7 |
| SICL (train-free) | 51.5 | 50.9 | 46.0 | 45.5 | 73.6 | 74.0 | 19.9 | 22.9 | 43.0 | 40.0 | 51.6 | 51.8 |
| SICL (train-driven) | **60.5** | **59.0** | **62.8** | **62.5** | **76.8** | **77.0** | **28.1** | **27.0** | **45.0** | **43.8** | **52.0** | **53.6** |

Quality Assessment (En–Ja, En–Zh, En–Cs); (4) Summarization (Gigaword); (5) Text Expansion (Gigatiny); (6) Machine Translation (WMT19 En–Zh). Task and dataset details are in Appendix Sections D–E. Evaluation metrics: Accuracy (classification), MSE (STS, TQA), ROUGE-1 (summarization, expansion), BLEU (translation). Higher Accuracy/ROUGE-1/BLEU or lower MSE indicates better performance. Metric computations are detailed in Appendix Section C.

**LLMs, baselines, training details and other setup:** SICL is evaluated on LLaMA 3.1 3B, LLaMA 3.1 8B Touvron et al. (2024), and Qwen2.5 7B Team (2024). Baseline methods include TTF (Liu et al., 2025), Delta-KNN (DKNN) (Li et al., 2025), KATE Liu et al. (2022b), Cluster-Diversity (CD) Naik et al. (2023), ICCL Liu et al. (2024b), and PPL (Gonen et al., 2023). **Training details**: We use BERT-base-uncased as the backbone encoder with six hidden layers. The model is trained for 10 epochs with a learning rate of $2 \times 10^{-5}$ and the AdamW optimizer. A dropout rate of 0.3 is applied to the classification layers. **Other setup**: All methods are evaluated under 5-shot and 10-shot demonstration settings. We adopt the T5 model (Raffel & Shazeer, 2025) as the pre-trained

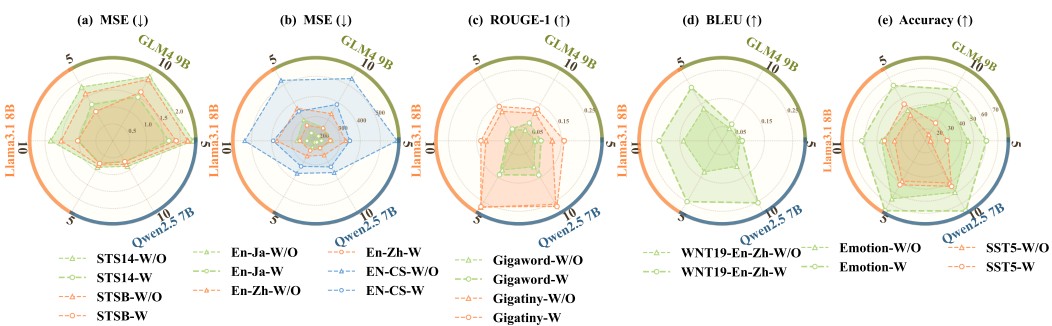

Figure 5: Experimental results with and without task difficulty simplification.

encoder (Section 3.1), with the maximum task width $w$ and depth $d$ fixed at 4 and 3, respectively. For ICL, the training and test sets of each dataset serve as the retrieval corpus and test set, respectively.

## 4.2 MAIN RESULTS

Table 1 reports the comparative results of SICL against baseline methods. SICL (train-driven) achieves SOTA performance across a range of tasks. In particular, for classification, it exceeds the next-best method by an average of 8.24% in accuracy; for STS, it reduces MSE by an average of 0.24; for TQA, it achieves an average decrease of 42.29 in MSE; for summarization and text expansion, it improves ROUGE-1 scores by 0.01 and 0.001, respectively; and for machine translation, it yields an average BLEU gain of 0.068. In addition, SICL (train-free) exhibits competitive performance, ranking second in most cases. This suggests that even relatively simple task simplification can lead to strong experimental performance.

## 4.3 ABLATION STUDY

(1) **The impact of task difficulty simplification:** We conduct an additional experiment in which SICL performs in-context learning with retrieved demonstrations but without simplifying the task. As shown in Fig. 5, the absence of simplification results in an average accuracy drop of 13.2% for classification tasks, an average degradation of 0.427 in MSE for STS, and an average degradation of 83.24 in MSE for TQA. By contrast, ROUGE-1 improves by an average of 0.019 for summarization and 0.01 for text expansion, while BLEU increases by 0.08 for machine translation. These results demonstrate that simplifying task difficulty substantially enhances SICL's performance. (2) **The impact of similarity-based retrieval:** Because SICL retrieves in-context examples based on similarity, we further examine the effect of this mechanism. Specifically, we replace the similarity-based retrieval module with retrieval strategies from BM25 and CD methods and evaluate SICL under this setting. As illustrated in Tab. 6, similarity-based retrieval consistently improves SICL's performance. For instance, on the STSB dataset with the Qwen2.5 7B LLM, SICL with similarity-based retrieval outperforms its counterpart without retrieval by 0.051 in BM25.

## 5 DISCUSSION AND LIMITATION

**The impact of task depth and width**: To investigate the effect of task depth and width on performance, we vary the maximum depth from 2 to 3. As shown in Tab. 2, increasing task depth initially enhances performance. For instance, on the STSB dataset with the Qwen2.5 7B model under the 5-shot setting, raising the maximum depth from 2 to 3 reduces the MSE from 0.783 to 0.765. Likewise, performance improves as task width increases from 2 to 3 and then stabilizes at 4. For example, on the STSB dataset with the Qwen2.5 7B model, the MSE decreases from 0.868 to 0.774 before leveling off at 0.765. **The impact of the number of in-context demonstrations:** We also examine the effect of varying the number of in-context demonstrations from two to ten (specifically, 2, 4, 6, 8, and 10). As shown in Fig. 6, although the model's performance fluctuates across different numbers of demonstrations, no consistent upward or downward trend is observed, indicating that demonstration count introduces largely random variation rather than a systematic effect.

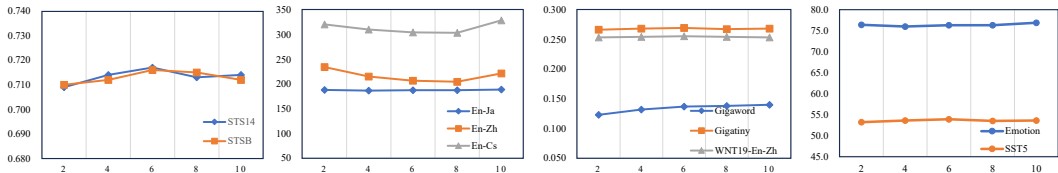

Figure 6: Results for different numbers of demonstrations. Similar performance is observed with a small number of demonstrations as with a large number.

Table 2: The results under different task depths and widths. Here, 2d and 3d indicate maximum task depths of 2 and 3, respectively, while 2w, 3w, and 4w denote maximum task widths of 2, 3, and 4, respectively.

| Metric | shot | data | 2d | 3d | 2w | 3w | 4w |
|---|---|---|---|---|---|---|---|
| MSE↓ | 5 | En-Ja | 201.521 | 189.606 | 220.500 | 200.670 | 189.606 |
| | | STSB | 0.843 | 0.834 | 0.934 | 0.844 | 0.834 |
| MSE↓ | 10 | En-Ja | 188.858 | 188.858 | 201.664 | 199.878 | 188.858 |
| | | STSB | 0.783 | 0.765 | 0.868 | 0.774 | 0.765 |
| Accuracy (%)↑ | 5 | Emotion | 73.1 | 76.8 | 70.2 | 75.4 | 76.8 |
| | | SST5 | 45.9 | 52.0 | 48.8 | 49.4 | 52.0 |
| Accuracy (%)↑ | 10 | Emotion | 76.3 | 77.0 | 70.9 | 74.1 | 77.0 |
| | | SST5 | 52.3 | 53.6 | 50.3 | 50.9 | 53.6 |

Table 3: The results of GPT-4o with SST5 dataset.

| BM25 | CD | Kate | DKNN | TTF | ICCL | PPL | SICL (train-driven) |
|---|---|---|---|---|---|---|---|
| 52.5 | 51.8 | 57.3 | 53.1 | 54.5 | 56.4 | 56.9 | 62.0 |

Table 4: Results of CIFAR 10 dataset.

| Metric | Accuracy | | |
|---|---|---|---|
| LLMs | VL26B | VL26B | VL26B |
| CR | 90.3 | 92.5 | 97.1 |
| CD | 85.7 | 90.9 | 97.5 |
| Kate | 92.8 | 92.5 | 98.6 |
| ICL-rerank | 93.5 | 90.8 | 98.1 |
| SICL (train-free) | 96.7 | 95.1 | 98.8 |

Table 5: The results of the plug-and-play integration.

| | With SICL | | Without SICL | |
|---|---|---|---|---|
| Method | Emotion | SST5 | Emotion | SST5 |
| CD | 52.5 | 48.2 | 39.8 | 45.9 |
| Kate | 47.4 | 53.1 | 40.0 | 51.8 |
| ICCL | 53.1 | 53.8 | 38.2 | 48.8 |
| PPL | 44.7 | 53.5 | 39.4 | 48.7 |

**Extension to commercial LLM**: SICL can be extended to commercial, closed-source LLMs such as GPT-4o. Using the SST5 dataset on GPT-4o, the results in Table 3 demonstrate that SICL (train-driven) SOTA performance, reaching an accuracy of 62.0%. **Application to vision domain**: SICL is also applicable to the vision domain. We evaluate it on the CIFAR 10 (Krizhevsky & Hinton, 2025) dataset using the multimodal LLM InternVL 2.5 (4B, 8B, and 26B). Under the configuration of 10 demonstrations with task depth and width fixed at 2 (i.e., the train-free SICL setting). Since methods like BM25 and PPL are only applicable to text, we supplement the comparison with Clustering-Retrieval (CR) Li & Qiu (2023a) and ICL-rerank. The results presented in Tab. 4 show that SICL achieves state-of-the-art (SOTA) performance, with an accuracy of 98.8%. **Plug-and-play integration with other ICL Methods**: SICL can function as a plug-and-play module to enhance existing ICL methods. We test this on the Emotion and SST5 datasets (Tab. 5), where SICL improves average of 8.95% accuracy. **Limitation:** SICL requires multiple LLM queries per input, increasing token and time costs. To further improve efficiency, we investigate reducing the number of demonstrations, as Section 5 shows their effect on performance is largely random. Fewer examples reduce the number of tokens fed into the LLM and shorten the amount of text the model must process, thereby lowering both token costs. Additionally, we employ multi-task querying in SICL (train-free) by simultaneously presenting two sub-questions to reduce time overhead. As shown in Tab. 7, when reducing demonstrations from 10 to 2 on the SST-5 dataset, accuracy decreases by only 0.4%, while average token usage drops to 57.4 tokens and inference time reduces to 0.35 seconds.

# 6 CONCLUSION

We introduce a new perspective on ICL that reduces task difficulty by progressively decomposing the output space. To this end, we develop two complementary approaches. The train-free variant of SICL is straightforward to implement and yields substantial performance gains, whereas the train-driven variant incurs additional training cost but achieves SOTA results. Furthermore, SICL generalizes well to a broad range of downstream tasks and demonstrates effectiveness even in the vision modality. Finally, we propose an efficiency-oriented strategy that reduces both token and time consumption, thereby further lowering SICL's computational overhead.

## ETHICS STATEMENT

This study adheres to the ICLR Code of Ethics. It does not involve human participants or animal experiments. All datasets used, including Emotion, SST5, Gigaword, Gigatiny, STS14, STSB, WNT19-En-Zh, and EN-CS—were obtained in compliance with their respective usage policies, ensuring no breaches of privacy. We took precautions to minimize potential biases and avoid discriminatory effects during the research. No personally identifiable information was accessed, and no aspects of the experiments posed privacy or security risks. We remain fully committed to transparency and integrity in our research practices.

## REPRODUCIBILITY STATEMENT

We have taken extensive steps to guarantee the reproducibility of our work. To facilitate verification and replication, both the source code and datasets will be released in an anonymous repository. The paper provides detailed descriptions of the experimental setup, covering training strategies and model configurations. For evaluation, we rely on publicly available datasets—Emotion, SST5, Gigaword, Gigatiny, STS14, STSB, WNT19-En-Zh, and EN-CS—ensuring fair and reproducible comparisons. These resources allow other researchers to validate our findings and further build upon them.

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

## OVERVIEW OF THE APPENDIX

This appendix includes our supplementary materials as follows:

- The LLM usage statement in Section A
- Experimental results in Section B
- More detailed descriptions of the evaluation metrics in Section C
- Formal definitions of the tasks in Section D
- More details about the datasets in Section E
- Models access information and URLs in Section F

## A LLM USAGE STATEMENT

Large Language Models (LLMs) were employed to assist in manuscript preparation by improving language quality, enhancing readability, and ensuring overall clarity. Their support was limited to tasks such as sentence refinement, grammatical corrections, and improving the flow of the text.

Importantly, LLMs did not contribute to the conceptualization, research methodology, or experimental design. All intellectual ideas, analytical approaches, and research content were entirely developed by the authors. LLM involvement was confined strictly to linguistic polishing, with no role in scientific analysis or content generation.

The authors take full responsibility for the manuscript, including the portions refined by LLMs, and affirm that all work adheres to ethical standards, with no plagiarism or misconduct.

## B EXPERIMENTAL RESULTS

Table 6: The ablation study compares different retrieval methods. BM25 denotes the use of the BM25 algorithm in SICL to obtain in-context examples. CD refers to applying the CD retrieval method within SICL for the same purpose. Similarity-based corresponds to the original SICL approach, which retrieves in-context examples based on similarity.

| LLMs | Llama3.2 3B | | Llama3.1 8B | | Qwen2.5 7B | | Llama3.2 3B | | Llama3.1 8B | | Qwen2.5 7B | |
|---|---|---|---|---|---|---|---|---|---|---|---|---|
| Shot | 5 | 10 | 5 | 10 | 5 | 10 | 5 | 10 | 5 | 10 | 5 | 10 |
| Metric | MSE ↓ | | | | | | MSE ↓ | | | | | |
| Task & Data | Semantic Textual Similarity: STS14 | | | | | | Semantic Textual Similarity: STSB | | | | | |
| BM25 | 1.982 | 1.776 | 1.317 | 1.274 | 0.870 | 0.771 | 1.891 | 1.746 | 1.191 | 1.105 | 0.959 | 0.886 |
| CD | 2.626 | 2.109 | 1.776 | 1.343 | 1.044 | 0.968 | 3.051 | 1.997 | 1.361 | 1.249 | 0.797 | 0.773 |
| Similarity-based | 1.657 | 1.491 | 1.239 | 1.018 | 0.771 | 0.714 | 1.872 | 1.663 | 0.996 | 1.035 | 0.776 | 0.712 |
| Metric | MSE ↓ | | | | | | MSE ↓ | | | | | |
| Task & Data | Translation Quality Assessment: En-Ja | | | | | | Translation Quality Assessment: En-Zh | | | | | |
| BM25 | 235.895 | 234.302 | 272.309 | 256.860 | 196.676 | 203.718 | 298.237 | 301.068 | 321.222 | 334.719 | 241.251 | 237.010 |
| CD | 248.455 | 244.102 | 267.435 | 245.651 | 197.648 | 198.276 | 301.843 | 287.799 | 292.985 | 306.654 | 215.347 | 230.424 |
| Similarity-based | 207.729 | 205.910 | 224.510 | 224.273 | 189.606 | 188.858 | 251.718 | 251.771 | 268.240 | 260.052 | 236.253 | 221.814 |
| Metric | MSE ↓ | | | | | | ROUGE-1 ↑ | | | | | |
| Task & Data | Translation Quality Assessment: EN-CS | | | | | | Summarization: Gigaword | | | | | |
| BM25 | 654.042 | 664.057 | 470.077 | 486.910 | 330.368 | 330.736 | 0.036 | 0.039 | 0.034 | 0.030 | 0.142 | 0.124 |
| CD | 403.008 | 423.116 | 400.216 | 533.333 | 329.766 | 327.055 | 0.071 | 0.052 | 0.055 | 0.046 | 0.152 | 0.151 |
| Similarity-based | 348.134 | 387.913 | 348.753 | 392.051 | 326.825 | 329.477 | 0.078 | 0.073 | 0.048 | 0.043 | 0.139 | 0.140 |
| Metric | ROUGE-1 ↑ | | | | | | BLEU ↑ | | | | | |
| Task & Data | Text expansion: Gigatiny | | | | | | Machine Translation: WNT19-En-Zh | | | | | |
| BM25 | 0.153 | 0.139 | 0.143 | 0.141 | 0.203 | 0.210 | 0.062 | 0.051 | 0.131 | 0.137 | 0.127 | 0.104 |
| CD | 0.134 | 0.117 | 0.142 | 0.139 | 0.282 | 0.271 | 0.067 | 0.052 | 0.055 | 0.046 | 0.152 | 0.151 |
| Similarity-based | 0.160 | 0.129 | 0.140 | 0.135 | 0.270 | 0.269 | 0.071 | 0.068 | 0.218 | 0.223 | 0.249 | 0.254 |
| Metric | Accuracy (%) ↑ | | | | | | Accuracy (%) ↑ | | | | | |
| Task & Data | Text classification: Emotion | | | | | | Text classification: SST5 | | | | | |
| BM25 | 51.7 | 53.3 | 49.8 | 51.6 | 69.1 | 69.9 | 24.6 | 25.3 | 40.2 | 38.0 | 51.2 | 52.7 |
| CD | 35.1 | 32.2 | 54.9 | 53.0 | 67.5 | 68.4 | 18.9 | 20.0 | 36.4 | 39.6 | 44.2 | 49.1 |
| Similarity-based | 60.5 | 59.0 | 62.8 | 62.5 | 76.8 | 77.0 | 28.1 | 27.0 | 45.0 | 43.8 | 52.0 | 53.6 |

## C METRICS

This appendix provides detailed definitions and computational procedures for the four evaluation metrics employed in our study: BLEU, ROUGE-1, MSE, and accuracy. These metrics are specifi-

Table 7: Results of time and token costs on the SST-5 dataset. Time = seconds per text; Token = tokens per text. SICL (train-free)-2m: train-free variant with multi-task prompting and 2 demonstrations. SICL (train-driven)-2: train-driven variant with 2 demonstrations.

| Method | BM25 | CD | Kate | DKNN | TTF | ICCL | PPL | SICL (train-free) | SICL (train-driven) | SICL (train-free)-2m | SICL (train-driven)-2 |
|---|---|---|---|---|---|---|---|---|---|---|---|
| Accuracy (%)↑ | 34.6 | 31.9 | 35.1 | 31.4 | 34.1 | 33.4 | 32.8 | 38.3 | 41.5 | 35.9 | 41.1 |
| Token↑ | 195.9 | 202.3 | 189.3 | 203.8 | 182.5 | 196.2 | 178.6 | 340.7 | 372.8 | 57.4 | 88.8 |
| Time (s)↓ | 1.95 | 0.26 | 0.27 | 0.22 | 0.27 | 0.36 | 0.32 | 0.43 | 0.51 | 0.35 | 0.48 |

cally chosen to assess different aspects of large language model performance across various natural language processing tasks.

## C.1 BLEU (BILINGUAL EVALUATION UNDERSTUDY)

BLEU is a precision-based metric primarily designed for evaluating machine translation quality by measuring the overlap between generated text and reference translations. The metric computes the geometric mean of modified n-gram precisions, typically considering n-grams up to length 4, combined with a brevity penalty to discourage overly short translations.

The BLEU score is calculated as follows:

$$\text{BLEU} = \text{BP} \times \exp\left(\sum_{n=1}^{N} w_n \log p_n\right) \tag{8}$$

where $p_n$ represents the modified n-gram precision:

$$p_n = \frac{\sum_{C \in \{Candidates\}} \sum_{n\text{-gram} \in C} \text{Count}_{\text{clip}}(n\text{-gram})}{\sum_{C' \in \{Candidates\}} \sum_{n\text{-gram}' \in C'} \text{Count}(n\text{-gram}')} \tag{9}$$

The brevity penalty (BP) is defined as:

$$\text{BP} = \begin{cases} 1 & \text{if } c > r \\ e^{(1-r/c)} & \text{if } c \leq r \end{cases} \tag{10}$$

where $c$ is the length of the candidate translation and $r$ is the effective reference corpus length. The weights $w_n$ are typically uniform ($w_n = 1/N$) for $N = 4$. Higher BLEU scores indicate better translation quality, with scores ranging from 0 to 1.

## C.2 ROUGE-1 (RECALL-ORIENTED UNDERSTUDY FOR GISTING EVALUATION)

ROUGE-1 is a recall-based metric specifically designed for evaluating automatic text summarization and text expansion tasks. Unlike BLEU's precision focus, ROUGE-1 emphasizes recall by measuring the proportion of unigrams in the reference text that appear in the generated text.

The ROUGE-1 score is computed as:

$$\text{ROUGE-1} = \frac{\sum_{s \in \{Reference\}} \sum_{gram_1 \in s} \text{Count}_{\text{match}}(gram_1)}{\sum_{s \in \{Reference\}} \sum_{gram_1 \in s} \text{Count}(gram_1)} \tag{11}$$

where $\text{Count}_{\text{match}}(gram_1)$ represents the maximum number of unigrams co-occurring in both the candidate and reference summaries, and $\text{Count}(gram_1)$ denotes the number of unigrams in the reference summary. ROUGE-1 can also be extended to compute precision and F1-measure:

$$\text{ROUGE-1}_{\text{precision}} = \frac{\sum_{s \in \{Candidate\}} \sum_{gram_1 \in s} \text{Count}_{\text{match}}(gram_1)}{\sum_{s \in \{Candidate\}} \sum_{gram_1 \in s} \text{Count}(gram_1)} \tag{12}$$

$$\text{ROUGE-1}_{\text{F1}} = \frac{2 \times \text{ROUGE-1}_{\text{precision}} \times \text{ROUGE-1}_{\text{recall}}}{\text{ROUGE-1}_{\text{precision}} + \text{ROUGE-1}_{\text{recall}}} \tag{13}$$

Higher ROUGE-1 scores indicate better content overlap between generated and reference texts, with scores ranging from 0 to 1.

## C.3 MSE (MEAN SQUARED ERROR)

MSE is a regression-based metric employed for evaluating semantic textual similarity (STS) tasks, where the objective is to predict continuous similarity scores between text pairs. MSE quantifies the average squared differences between predicted and ground-truth similarity scores.

The MSE is calculated as:

$$\text{MSE} = \frac{1}{n} \sum_{i=1}^{n} (y_i - \hat{y}_i)^2 \tag{14}$$

where $n$ represents the total number of text pairs, $y_i$ denotes the ground-truth similarity score for the $i$-th text pair, and $\hat{y}_i$ represents the predicted similarity score. In the context of semantic textual similarity, similarity scores are typically normalized to a continuous scale (e.g., 0 to 5 or 0 to 1).

Lower MSE values indicate better model performance, with MSE = 0 representing perfect prediction accuracy. The metric is particularly sensitive to outliers due to the squaring operation, making it effective for penalizing large prediction errors in similarity assessment tasks.

## C.4 ACCURACY

Accuracy is a classification-based metric used for evaluating discrete text classification tasks, measuring the proportion of correctly classified instances out of the total number of instances. This metric is particularly suitable for tasks with well-defined categorical outputs.

The accuracy is computed as:

$$\text{Accuracy} = \frac{\text{Number of Correct Predictions}}{\text{Total Number of Predictions}} = \frac{1}{n} \sum_{i=1}^{n} \mathbf{1}(y_i = \hat{y}_i) \tag{15}$$

where $n$ is the total number of test instances, $y_i$ represents the true class label for the $i$-th instance, $\hat{y}_i$ denotes the predicted class label, and $\mathbf{1}(\cdot)$ is the indicator function that returns 1 if the condition is true and 0 otherwise.

For multi-class classification problems, accuracy can be further decomposed into class-specific accuracies or complemented with other metrics such as precision, recall, and F1-score to provide a more comprehensive evaluation. Higher accuracy scores indicate better classification performance, with scores ranging from 0 to 1, where 1 represents perfect classification.

In summary, these four metrics collectively provide a comprehensive evaluation framework for assessing large language model performance across diverse natural language processing tasks: BLEU for translation quality, ROUGE-1 for summarization effectiveness, MSE for similarity prediction accuracy, and accuracy for classification performance.

# D  TASK

This appendix provides comprehensive definitions and descriptions of the five fundamental natural language processing tasks employed in our evaluation framework. These tasks represent diverse aspects of language understanding and generation capabilities, enabling a thorough assessment of large language model performance across different linguistic competencies.

## D.1  TEXT CLASSIFICATION

Text classification is a supervised learning task that involves assigning predefined categorical labels to textual inputs based on their semantic content, stylistic features, or contextual characteristics. This task encompasses a broad spectrum of applications, ranging from binary sentiment analysis to multi-class topic categorization and hierarchical document classification.

Formally, given a text document $d$ represented as a sequence of tokens $d = \{w_1, w_2, \ldots, w_n\}$ and a finite set of predefined categories $C = \{c_1, c_2, \ldots, c_k\}$, the objective is to learn a mapping function $f : D \to C$ that assigns the most appropriate class label to each document. The classification

process typically involves feature extraction from the input text, followed by the application of machine learning algorithms to predict the target category.

Text classification serves as a fundamental benchmark for evaluating model comprehension capabilities, as it requires the system to understand semantic nuances, contextual relationships, and domain-specific terminology. Common applications include spam detection, news categorization, product review classification, and intent recognition in conversational systems.

### D.2  TEXTUAL SIMILARITY ESTIMATION

Textual similarity estimation, also known as semantic textual similarity (STS), is a regression task that quantifies the degree of semantic relatedness between pairs of textual inputs. Unlike binary similarity judgments, this task requires models to produce continuous similarity scores that reflect fine-grained semantic relationships.

Given two text segments $s_1$ and $s_2$, the goal is to predict a similarity score $sim(s_1, s_2) \in [0, 1]$ or $sim(s_1, s_2) \in [0, 5]$, depending on the evaluation framework, where higher scores indicate greater semantic similarity. The task demands sophisticated understanding of lexical semantics, syntactic variations, and contextual meaning preservation across different linguistic expressions.

Textual similarity estimation challenges models to capture various dimensions of semantic relatedness, including synonymy, paraphrase detection, textual entailment, and conceptual overlap. This task is particularly valuable for assessing a model's ability to understand meaning beyond surface-level lexical matching, requiring deep comprehension of semantic composition and contextual inference. Applications include information retrieval, duplicate detection, plagiarism identification, and question-answering systems.

### D.3  ABSTRACTIVE SUMMARIZATION

Abstractive summarization is a text generation task that involves creating concise, coherent summaries by synthesizing and reformulating information from source documents rather than merely extracting existing sentences. This task requires sophisticated natural language understanding and generation capabilities, as models must comprehend the source content, identify salient information, and produce novel textual expressions.

The task can be formally defined as learning a function $g : D \rightarrow S$, where $D$ represents the input document(s) and $S$ denotes the generated summary. Unlike extractive summarization, which selects and concatenates existing sentences, abstractive summarization demands the generation of new sentences that capture the essential meaning while maintaining factual accuracy and linguistic fluency.

Abstractive summarization evaluation encompasses multiple dimensions, including content preservation, factual consistency, linguistic quality, and compression ratio. The task requires models to demonstrate advanced capabilities in information synthesis, semantic abstraction, and coherent text generation. This makes it an excellent benchmark for assessing both comprehension and generation abilities in large language models. Applications span document summarization, news article condensation, scientific paper abstracts, and meeting transcript summarization.

### D.4  TEXT EXPANSION

Text expansion is a generative task that involves elaborating, extending, or providing additional context to given textual inputs while maintaining semantic coherence and topical relevance. This task requires models to demonstrate creative language generation capabilities while adhering to the thematic and stylistic constraints established by the input text.

The objective is to learn a mapping function $h : T_{short} \rightarrow T_{expanded}$, where $T_{short}$ represents the input text segment and $T_{expanded}$ denotes the augmented output that preserves the original meaning while providing additional detail, explanation, or context. The expansion process may involve various linguistic operations, including elaboration of concepts, provision of examples, addition of supporting details, or enhancement of descriptive elements.

Text expansion evaluation typically focuses on semantic consistency, relevance maintenance, factual accuracy, and linguistic fluency of the generated content. The task serves as an effective measure of a model's ability to generate coherent, contextually appropriate text while maintaining thematic unity. This capability is essential for applications such as content augmentation, educational material development, creative writing assistance, and detailed explanation generation.

### D.5 TRANSLATION

Machine translation is a sequence-to-sequence task that involves converting text from a source language to a target language while preserving semantic meaning, stylistic nuances, and contextual appropriateness. This task represents one of the most challenging applications in natural language processing, requiring comprehensive understanding of linguistic structures, cultural contexts, and cross-lingual semantic correspondences.

Formally, given a source sentence $S = \{s_1, s_2, \ldots, s_m\}$ in language $L_s$ and a target language $L_t$, the goal is to generate a translation $T = \{t_1, t_2, \ldots, t_n\}$ that accurately conveys the meaning of $S$ in $L_t$. Modern neural machine translation approaches typically employ encoder-decoder architectures with attention mechanisms to capture long-range dependencies and align source and target language elements.

Translation quality assessment encompasses multiple criteria, including adequacy (semantic preservation), fluency (linguistic naturalness), and faithfulness (factual consistency). The task requires models to navigate complex linguistic phenomena such as syntactic divergence, lexical ambiguity, idiomatic expressions, and cultural references. Translation serves as a comprehensive evaluation benchmark that tests both multilingual understanding and cross-lingual generation capabilities, making it invaluable for assessing the global applicability and linguistic versatility of large language models.

In summary, these five tasks collectively provide a comprehensive evaluation framework that assesses different aspects of natural language processing capabilities: classification for understanding, similarity estimation for semantic reasoning, summarization for information synthesis, expansion for creative generation, and translation for cross-lingual competence.

## E DATASET

This appendix provides comprehensive descriptions of the nine datasets employed in our experimental evaluation. These datasets represent diverse natural language processing tasks and domains, enabling thorough assessment of model performance across different linguistic challenges and application scenarios.

### E.1 SST5 (STANFORD SENTIMENT TREEBANK-5)

The Stanford Sentiment Treebank-5 (SST5) is a fine-grained sentiment analysis dataset that extends the original binary Stanford Sentiment Treebank with five sentiment categories: very negative, negative, neutral, positive, and very positive. The dataset comprises movie reviews from Rotten Tomatoes, with sentiment labels assigned not only to complete sentences but also to sub-phrases and individual words, creating a hierarchical sentiment structure.

SST5 contains approximately 11,855 sentences with fine-grained sentiment annotations, making it a challenging benchmark for sentiment classification tasks. The dataset's hierarchical structure provides compositional sentiment information, allowing models to understand how sentiment is constructed from individual components. The five-class classification scheme requires models to distinguish subtle sentiment gradations beyond simple positive/negative categorization, testing their ability to capture nuanced emotional expressions and contextual sentiment variations.

### E.2 EMOTION

The Emotion dataset is a multi-class text classification benchmark designed for emotion recognition in textual content. This dataset categorizes text into six basic emotion categories based on Ekman's

Table 8: Prompts used for different datasets

| Dataset | Prompt |
|---------|--------|
| Emotion | You are a helpful assistant. Predict the label of the input text, only give me the label is enough, for instance, label = 'Anger', label = 'Fear', label = 'Joy', label = 'Love', label = 'Sadness', label = 'Surprise'. Labels are Anger, Fear, Joy, Love, Sadness, and Surprise, not other labels. |
| SST5 | You are a helpful assistant. Predict the label of the input text, only give me the label is enough, for instance, label = 'very negative', label = 'negative', label = 'neutral', label = 'positive', label = 'very positive'. |
| SST14 | You are a helpful assistant. You are asked to predict the semantic textual similarity of every input text pairs. Your response only contains a single numerical value with the range from 0 to 5. A larger number indicates a higher degree of similarity. |
| STSB | You are a helpful assistant. You are asked to predict the semantic textual similarity of every input text pairs. Your response only contains a single numerical value with the range from 0 to 5. A larger number indicates a higher degree of similarity. |
| gigatiny | You are a helpful assistant. Expand this paragraph without altering its core meaning. |
| gigaword | You are a helpful assistant. Summarize the following text and generate an abstract. |
| wmt19_Zh-En | You are a helpful assistant. Translate the following text from Chinese to English. |

fundamental emotions: anger, disgust, fear, joy, sadness, and surprise. The dataset consists of English Twitter messages that have been manually annotated with emotion labels.

The dataset contains approximately 20,000 training examples and 2,000 test examples, providing a substantial corpus for emotion classification tasks. Each text sample is associated with a single dominant emotion label, requiring models to understand subtle linguistic cues, contextual implications, and emotional expressions in informal social media text. The dataset presents unique challenges due to the informal nature of Twitter language, including abbreviations, emoticons, hashtags, and colloquial expressions that require sophisticated natural language understanding capabilities.

### E.3 STS14 (SEMANTIC TEXTUAL SIMILARITY 2014)

STS14 is part of the SemEval-2014 Semantic Textual Similarity shared task, designed to evaluate systems' ability to assess semantic similarity between sentence pairs. The dataset comprises sentence pairs drawn from multiple domains, including news headlines, image captions, forum discussions, and WordNet definitions. Each sentence pair is annotated with a continuous similarity score ranging from 0 (completely dissimilar) to 5 (semantically equivalent).

The dataset contains approximately 3,750 sentence pairs across six different domains, providing diverse linguistic contexts and semantic relationships. STS14 challenges models to understand various dimensions of semantic similarity, including lexical overlap, syntactic variation, semantic composition, and contextual meaning. The multi-domain nature of the dataset tests model robustness across different text types and linguistic styles, making it a comprehensive benchmark for semantic similarity estimation.

### E.4 STS15 (SEMANTIC TEXTUAL SIMILARITY 2015)

STS15 continues the SemEval semantic textual similarity evaluation series, featuring sentence pairs from diverse sources including news articles, image captions, student answers, and belief statements. The dataset maintains the same 0-5 similarity scoring scheme as previous STS benchmarks, providing continuous similarity assessments rather than binary judgments.

The dataset consists of approximately 3,000 sentence pairs distributed across five domains, each presenting unique linguistic challenges and semantic relationship types. STS15 introduces additional

complexity through domain-specific terminology and varied text lengths, requiring models to handle both short phrases and longer sentences effectively. The dataset's emphasis on cross-domain generalization makes it particularly valuable for assessing model robustness and semantic understanding capabilities across different textual contexts.

### E.5 STS16 (SEMANTIC TEXTUAL SIMILARITY 2016)

STS16 represents the final iteration of the SemEval semantic textual similarity shared task series, incorporating lessons learned from previous years while introducing new challenges. The dataset includes sentence pairs from news headlines, plagiarism detection, post-editing, question-question similarity, and answer-answer similarity domains.

Containing approximately 1,379 sentence pairs across five domains, STS16 presents diverse semantic similarity challenges, including paraphrase detection, textual entailment recognition, and semantic equivalence assessment. The dataset's inclusion of question-answer pairs and plagiarism detection scenarios tests models' ability to understand semantic relationships in specialized contexts. The varied domain distribution ensures comprehensive evaluation of semantic similarity estimation across different application scenarios.

### E.6 STSB (SEMANTIC TEXTUAL SIMILARITY BENCHMARK)

The Semantic Textual Similarity Benchmark (STSB) is a comprehensive dataset that consolidates and standardizes semantic textual similarity evaluation. STSB combines sentence pairs from previous STS shared tasks (2012-2017) with additional annotations, creating a unified benchmark for semantic similarity assessment.

The dataset comprises approximately 8,628 sentence pairs with human-annotated similarity scores on a 0-5 scale, providing a substantial corpus for both training and evaluation. STSB includes diverse text types ranging from news headlines and image captions to forum posts and video descriptions. The dataset's standardized format and comprehensive coverage make it a widely adopted benchmark for evaluating semantic textual similarity systems, offering reliable performance comparisons across different approaches and model architectures.

### E.7 GIGAWORD

The Gigaword dataset is a large-scale abstractive summarization corpus derived from the Annotated Gigaword Fifth Edition. The dataset consists of article headline generation tasks, where models must produce concise, informative headlines from the first sentence of news articles. This formulation transforms headline generation into an abstractive summarization task requiring content compression and reformulation.

The dataset contains approximately 3.8 million training examples and 189,651 test examples, making it one of the largest available summarization benchmarks. Each example pairs a news article's first sentence with its corresponding headline, requiring models to identify key information and generate concise summaries. The dataset's large scale enables training of data-intensive neural models while its news domain focus provides consistent linguistic style and content structure.

### E.8 GIGATINY

Gigatiny is a reduced version of the Gigaword dataset, designed to provide a more manageable benchmark for abstractive summarization evaluation while maintaining the essential characteristics of the original corpus. This subset maintains the same task formulation of generating headlines from article first sentences but with significantly reduced data volume.

The dataset typically contains approximately 100,000 training examples, offering a balance between computational efficiency and sufficient data for model training. Gigatiny preserves the linguistic diversity and summarization challenges of the full Gigaword dataset while enabling faster experimentation and evaluation cycles. The reduced size makes it particularly suitable for comparative studies and ablation experiments where computational resources are limited.

### E.9  WMT19-EN-ZH (WMT 2019 ENGLISH-CHINESE)

WMT19-EN-ZH is the English-to-Chinese translation dataset from the Fourth Conference on Machine Translation (WMT19) shared task. This dataset represents one of the most challenging language pairs in machine translation due to significant linguistic differences between English and Chinese, including different writing systems, syntactic structures, and cultural contexts.

The dataset contains approximately 25 million parallel sentence pairs for training, with additional development and test sets for evaluation. The corpus includes diverse domains such as news, government documents, and web content, providing comprehensive coverage of translation scenarios. WMT19-EN-ZH presents unique challenges including character-based writing systems, different word segmentation approaches, syntactic divergence, and cultural reference translation. The dataset serves as a rigorous benchmark for assessing cross-lingual understanding and generation capabilities in neural machine translation systems.

These nine datasets collectively provide a comprehensive evaluation framework spanning text classification, semantic similarity estimation, abstractive summarization, and machine translation tasks, enabling thorough assessment of large language model capabilities across diverse natural language processing challenges.

Table 9: Datasets and Their URLs

| Task | Dataset | URL |
|------|---------|-----|
| Abstractive Summarization / Text Expansion | gigaword | `https://huggingface.co/datasets/Gabriel/gigaword_swe` |
| | gigatiny | `https://huggingface.co/datasets/SpeedOfMagic/gigaword_tiny` |
| Text Classification | SST5 | `https://huggingface.co/datasets/SetFit/sst5` |
| | Emotion | `https://huggingface.co/datasets/dair-ai/emotion` |
| Translation | WMT19 En–Zh | `https://huggingface.co/datasets/WillHeld/wmt19-valid-only-zh_en` |
| Textual Similarity Estimation | STS14 | `https://huggingface.co/datasets/mteb/sts14-sts` |
| | STSB | `https://huggingface.co/datasets/SetFit/stsb` |
| | STS15 | `https://huggingface.co/datasets/mteb/sts15-sts` |
| | STS16 | `https://huggingface.co/datasets/mteb/sts16-sts` |

## F  THE URL OF MODELS

Table 10: Large Language Models and Their URLs

| Model | URL |
|-------|-----|
| GPT-4o | `https://platform.openai.com/docs/models/gpt-4o` |
| LLAMA-3.2-3b | `https://huggingface.co/meta-llama/Llama-3.2-3B` |
| GLM4 9B | `https://huggingface.co/zai-org/glm-4-9b` |
| Qwen2.5 7b | `https://huggingface.co/Qwen/Qwen2.5-7B` |
| LLAMA-3.1-8b | `https://huggingface.co/meta-llama/Llama-3.1-8B` |

