# OpenReview forum: "Simplify In-Context Learning"
_ICLR.cc/2026/Conference — ICLR 2026 Conference Withdrawn Submission_

### Official Review · Reviewer_NkrK · 2025-10-27

**Soundness:** 2
**Presentation:** 2
**Contribution:** 2
**Rating:** 2
**Confidence:** 3

**Summary:**

This paper proposes simplified in-context learning. This key motivation is to reduce tasks difficulty by decomposing the label space into easier subtasks with two dimensions: width is labels per subtask and depth is the number of subtasks. While their method is applied on large language model with ICL, they introduce two variant. The first is a training-free split by using for example clustering and the second is to train a score predictor to choose per-query partitions. The method is also extended to generation tasks by sampleing top-k candidates and picking one with a learned scorer.  The experimental results are reported on classification such as STS and generation such as summarization and translation, which show improved performance over previous ICL sample selection methods.

**Strengths:**

1. The motivation of this paper is intuitive and different, which study on output space simplification instead of widely focused demo selection for ICL.
2. The decomposition mechanics such as width/depth and score predictor are well specified.
2. The proposed training-free and training-driven modes are practical from quick to adaptive, the experimental results verifiy them as well.

**Weaknesses:**

1. It sounds the motivation of this paper is overclaim without enough evidence. The paper makes a premise that better in-contexts become ineffective when task difficulty exceeds models' capability. I don't see "task difficulty exceeds models' capability" where the experiments are only conducted on tasks (summarization, translation, etc) that the models can handle well.
2. The paper claims an urgent, neglected space of “task-difficulty reduction,” but decomposition-based prompting is well established, such as Least-to-Most Prompting, Decomposed Prompting and so on. They all target on making problem decompostion and make reasoning steps easier. The method proposed in this work is instead seem more restricted.
3. For generation extension, it is top-k sampliing with a learned reranker, which is a standard reranking setup and departs from the paper’s main contribution.
4. In Table 1, absolute scores for generation are extremely low (e.g., Gigaword ROUGE-1 ≈ 0.14; WMT19 En–Zh BLEU ≈ 0.25), which makes “SOTA” claims within the paper’s baseline suite less compelling.
5. The paper equates “difficulty” mostly to output-space size, and assumes subtasks are always easier, which may not hold when subtasks propagate early errors. This paper lacks robustness studies for this potential issue.

**Questions:**

See weaknesses.

---

### Official Review · Reviewer_CCZ6 · 2025-11-01

**Soundness:** 1
**Presentation:** 2
**Contribution:** 1
**Rating:** 2
**Confidence:** 5

**Summary:**

This paper introduces Simplified In-context Learning (SICL), a framework designed to reduce task complexity by decomposing a challenging task into a series of simpler subtasks. SICL combines two complementary approaches: training-free methods, which quickly decompose tasks by clustering or uniformly partitioning the output space, and training-based methods, which use a scoring predictor to adaptively identify the optimal decomposition for each query.

**Strengths:**

This paper introduces Simplified In-context Learning (SICL), a framework designed to reduce task complexity by decomposing a challenging task into a series of simpler subtasks, which shows performance improvements compared to baselines such as BM25.

**Weaknesses:**

1. Although the authors claim that "*Current ICL emphasizes enhancing LLM capability via in-context adjustments but neglects task difficulty reduction,*" the concept of decomposing a complex task into simpler subtasks is not original. Similar ideas have been extensively explored in prior work, such as plan-and-solve prompting [1]. The authors should conduct a more comprehensive literature review of related methods instead of asserting that this area remains underexplored. Furthermore, the paper lacks a thorough evaluation of the quality and coherence of the decomposed subtasks.
2. The proposed approach is limited to simple regression and classification tasks. While the authors claim that the method can be generalized to generation tasks, the generation-based method has a significant gap compared to its classification/regression counterparts, which resembles a straightforward best-of-N strategy rather than the proposed new method.
3. The experimental comparison is not comprehensive, as several state-of-the-art prompting and in-context learning techniques are missing from the evaluation. A broader experimental comparison would strengthen the validity of the paper's conclusions, especially, the "state-of-the-art."
4. Paper presentation issues:
    - Citation issues: For example, Qwen2.5 7B Team (2024) (`\citet{}`) $\rightarrow$ `\citep{}`
    - Page 4: $ymin$ $\rightarrow$ $y_{min}$

**References:**

[1] Plan-and-Solve Prompting: Improving Zero-Shot Chain-of-Thought Reasoning by Large Language Models (Wang et al., ACL 2023)

**Questions:**

See "Weaknesses."

---

### Official Review · Reviewer_Ts1D · 2025-11-01

**Soundness:** 2
**Presentation:** 2
**Contribution:** 2
**Rating:** 2
**Confidence:** 4

**Summary:**

The paper argues that many ICL methods try to raise a model’s capability via better retrieval/formatting/ordering of demonstrations, whereas performance is often bottlenecked by the task’s difficulty. It proposes Simplified In‑Context Learning (SICL), which reduces task difficulty by decomposing a task into simpler subtasks whose output spaces are progressively constrained. SICL has two flavors: (i) a training‑free version that uses K‑means (classification) or mid‑point splitting (regression) with fixed width/depth, and (ii) a training‑driven version that learns a Score Predictor (SP) to adaptively choose the decomposition (width and depth) per query. The authors report SOTA across six tasks, three LLMs, and ten datasets; they also extend to generative tasks and to a vision benchmark.

**Strengths:**

- The paper introduces SICL (Simplified In-Context Learning), a novel framework that simplifies in-context learning by decomposing complex tasks into simpler, more manageable subtasks. It proposes two complementary approaches: a train-free method, which rapidly partitions tasks without additional training, and a train-driven method, which adaptively learns optimal decompositions for improved task performance.

- Extensive experiments covering various datasets (classification, regression, summarization, machine translation, etc.) and multiple LLMs (LLaMA, Qwen) showcase broad applicability and effectiveness.

**Weaknesses:**

- The paper is somewhat challenging to follow. Section 3, in particular, includes complex notation and unnecessary mathematical formulations, making key concepts harder to grasp.
- The train-free method is limited to classification and regression tasks, restricting its applicability to generative and other NLP tasks.
- It is initially unclear how SICL operates during inference. Specifically, whether each demonstration includes subtasks or if the model separately solves subtasks sequentially. Only towards the end does it become apparent that subtasks are sequentially solved. An explicit illustrative example earlier in the paper would significantly improve clarity.
- Evaluations are conducted on tasks with relatively small label spaces. The effectiveness of SICL on larger-scale tasks (with significantly more labels) is not explored. As the label space grows, the efficiency of SICL could diminish substantially due to increased subtasks.
- The paper does not thoroughly analyze the specific sources of performance gain provided by SICL. It remains unclear how much improvement stems directly from solving simpler subtasks versus seeing additional demonstrations or repeated demonstrations across subtasks.
- The motivation behind the train-driven approach is not sufficiently justified. Achieving state-of-the-art results is less surprising given this method involves explicit training. Clarifying the motivations of the train-driven approach compared to fully train-free methods would strengthen the paper.
- The claim that the framework "demonstrates SOTA performance on proprietary commercial models" might be overstated, as only one dataset (SST5) is evaluated using GPT-4o on just the train-driven method.

**Questions:**

1. How many training examples are required for the train-driven SICL method to achieve effective performance?
2. What decoding strategy are you using? What temperature setting was used during inference in your experiments, and how many inference runs were conducted per evaluation?
3. How does your approach fundamentally differ from chain-of-thought (CoT) prompting? All subtasks could be interpreted as intermediate reasoning steps, so clarifying this distinction would be helpful.
4. What is GPT-4o's train-free SICL performance on SST5? Additionally, could you report GPT-4o's performance on regression and generation tasks for comparison?

---

### Official Review · Reviewer_Xfsu · 2025-11-03

**Soundness:** 3
**Presentation:** 3
**Contribution:** 2
**Rating:** 6
**Confidence:** 3

**Summary:**

The paper  proposes a framework called Simplified In-context Learning (SICL), which aims to reduce the difficulty of a task rather than (which is more common in the literature) only improving an LLM capability. SICL proposes to achieve this by decomposing complex tasks into simpler subtasks. The proposed approach sequentially constrains the output space to binary or smaller-range decisions. To implement this concept, the paper proposes two strategies. The first is a training-free method that uses fixed decomposition structure. The second training-driven method that adaptively determines the optimal decomposition for each query using a score predictor. Empirical results across six tasks, three LLMs, and ten datasets are provided.

**Strengths:**

*Perspective shift. The paper provides and interesting emphasis on reducing task difficulty rather than improving model capability, which is conceptually intuitive and refreshing.
*Decomposition framework. Although the idea of task decomposition is not entirely new (e.g. chain-of-thought), the authors systematic application of ICL is relatively new.
*Two modes. The paper offers both training-free and training-driven variants, which is a nice touch showing practical consideration.

**Weaknesses:**

*Fundamental assumption. The paper highlights that coarse-grained labels are 'inherently easier' but no empirical evidence is presented. Does predicting 'dog' vs 'Corgi' actually improve the accuracy of LLMs independent of output space size? This feels more like an assumption rather than proven fact.
*Theoretical foundations/difficulty level: The paper seem to lack theoretical analysis when, why or under what conditions the decomposition leads to reduction of difficulty. For example, why predicting within [0,50] should be easier than [0,100] for an LLM? Is it because of reduced decision space complexity? Narrower ranges of LLM probabilities?
*Error Analysis. When dealing with hierarchical decomposition, one can find that it creates a chain of decision where errors can compound, however, this was not discussed or analysed. For example, if the first subtask makes the wrong decision, all predictions that follow will be wrong. How often this happens?

**Questions:**

1. Can you explain what percentage of SICL errors are due to mistakes at each subtask level, and how do error patterns compare to baseline methods? As mentioned before, this is to understand the risk error compounding. If you show that (a) subtask 1 accuracy is very high or (b) even with some subtask 1 errors, subsequent preditions can  ‘recover’, this would potentially easy the concern about small error snowballing.
2. Why is K-means on label embeddings the right way to decompose label spaces. Also, how sensitive are results to clustering method and it’s quality?
3. How much training data does the score predictor need, and can it generalise across domains/LLMs?
4. Are the reported improvements statistically significant, and how stable are results across different random seeds and demonstration samples? Currently no error bars or confidence intervals are given in the tables. Also, I couldn’t find the discussion of multiple runs with different seeds. At this point it is unclear whether improvements like 0.776 versus 0.712 MSE are significant.
5. Could you characterise or provide some informed discussion when SICL helps or when it doesn't, based on task properties?

---

### Note · Authors · 2026-01-05

I have read and agree with the venue's withdrawal policy on behalf of myself and my co-authors.